# Towards Low-latency Event-based Visual Recognition with Hybrid Step-wise Distillation Spiking Neural Networks

## ABSTRACT

Spiking neural networks (SNNs) have garnered significant attention for their low power consumption and high biological interpretability. Their rich spatio-temporal information processing capability and event-driven nature make them ideally well-suited for neuromorphic datasets. However, current SNNs struggle to balance accuracy and latency in classifying these datasets. In this paper, we propose Hybrid Step-wise Distillation (HSD) method, tailored for neuromorphic datasets, to mitigate the notable decline in performance at lower time steps. Our work disentangles the dependency between the number of event frames and the time steps of SNNs, utilizing more event frames during the training stage to improve performance, while using fewer event frames during the inference stage to reduce latency. Nevertheless, the average output of SNNs across all time steps is susceptible to individual time step with abnormal outputs, particularly at extremely low time steps. To tackle this issue, we implement Step-wise Knowledge Distillation (SKD) module that considers variations in the output distribution of SNNs at each time step. Empirical evidence demonstrates that our method yields competitive performance in classification tasks on neuromorphic datasets, especially at lower time steps.

## CCS CONCEPTS

• **Computing methodologies → Computer vision**; **Neural networks**.

## KEYWORDS

Spiking Neural Networks, Neuromorphic Vision, Low-Latency, Hybrid Training, Knowledge Distillation

## 1 INTRODUCTION

In recent years, spiking neural networks (SNNs) [26], heralded as the next generation neural network paradigm, have garnered considerable attention due to their rich spatio-temporal characteristics and event-driven communication [31]. Notably, compared to conventional neural networks, SNNs exhibit the distinctive advantage of low power consumption.

Indeed, SNN manifests a significant advantage in terms of low power consumption due to their transmission of information through binary 0 or 1 signals [6]. However, satisfactory results are achieved by accumulating information over multiple time steps, leading to

*ACM MM, 2024, Melbourne, Australia*
© 2024 Copyright held by the owner/author(s). Publication rights licensed to ACM.
ACM ISBN 978-x-xxxx-xxxx-x/YY/MM
https://doi.org/10.1145/nnnnnnn.nnnnnnn

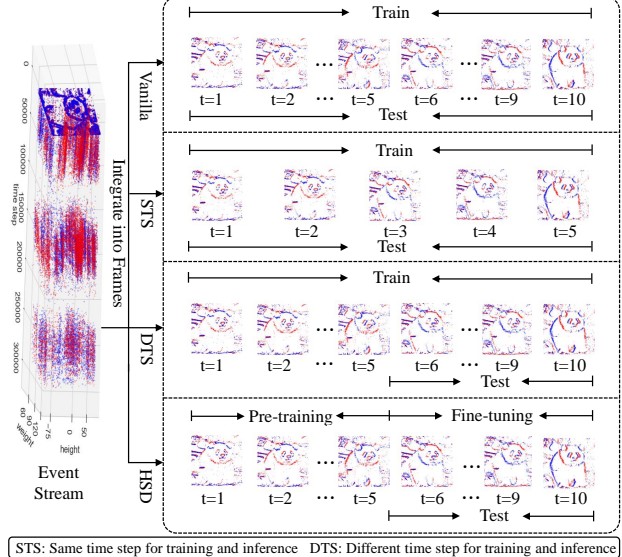

**Figure 1: Contrast of Vanilla, STS, DTS, and HSD methods. Both the training and inference stages in the Vanilla method utilize a uniform time step, often set to 10. STS maintains this time step solely during inference to minimize duration. DTS and HSD both adopt variable time steps, with HSD further segmenting the SNN training stage.**

heightened inference latency and increased power consumption [6]. To strike a balance between accuracy and latency, two primary methods for training SNNs have been proposed: the artificial neural network (ANN)-SNN conversion method [2, 3, 8, 9, 32] and the direct training SNN method [8, 12, 44, 46]. The former achieves high accuracy after converting weights from a pre-trained ANN but experiences noticeable latency during inference stage. The utilization of surrogate gradients during SNN training via back-propagation reduces latency but comes with the drawback of information transmission loss [15]. In particular, due to the temporal dimension inherent in the spiking neurons of SNNs, it renders them exceptionally well-suited for neuromorphic, event data [39]. The event data captured by event cameras record variations in light intensity at each pixel, resulting in a sequence of events that include the pixel's location, time, and polarity arising from changes in light intensity.

On the one hand, the reductions in latency are primarily observable in traditional static data scenarios, *e.g.*, CIFAR 10/100 and even the larger ImageNet dataset [3, 8]. And in other event-based vision tasks, such as dense prediction, recent works have demonstrated high accuracy can be achieved at extremely low time steps in the inference stage [4, 16]. This motivates us to consider a similar thought, which naturally raises the question: *How to achieve a trade-off between high accuracy and low-latency in event-based visual recognition?*

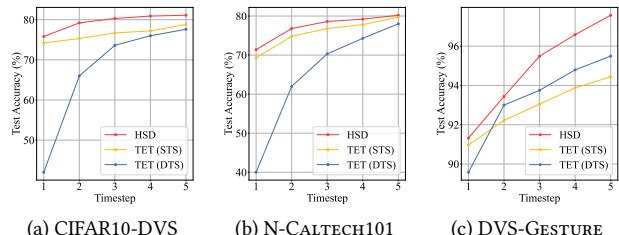

(a) CIFAR10-DVS     (b) N-Caltech101     (c) DVS-Gesture

**Figure 2: Comparisons performances of TET with STS, TET with DTS and HSD at time step $T$ = 1 to 5 on CIFAR10-DVS, N-Caltech101, and DVS-Gesture.**

The existing methods require higher latency for SNNs to recognize event-based neuromorphic objects, *e.g.*, CIFAR10-DVS [22], N-CALTECH101 [28], and DVS-GESTURE [1], which still require 10 time steps. As illustrated in Fig. 1, to reduce the latency in the inference stage, we believe that it can be divided into STS and DTS, where STS representing ensuring the consistency of time steps between the training and inference stages of SNNs, thereby reducing the overall time step during SNN training, DTS does not guarantee the consistency of time steps between the training and inference stages of SNNs, it only decreases the time step during the inference stage. And as shown in Fig. 2, when the time step is reduced, especially when the time step is less than or equal to 5, there is a significant decrease in performance. This decrease in accuracy can be mainly attributed to the sparsity of event data and the limited accumulation of event frame information in lower time steps [13, 42, 43]. Furthermore, we observe that STS often outperforms DTS, particularly at lower time steps. Through analysis, we attribute the performance gap to the fact that using an equal number of event frames for both training and inference allows the SNN to capture the dynamic features and contextual information of the complete event data learned during training. In contrast, inconsistency results in information loss, leading to a significant performance drop.

Hence, achieving higher performance often requires maintaining consistency in the time steps between the training and inference stages of SNNs, but this is closely associated with increased latency. We identify the sparse nature of event data as the primary cause of this issue, necessitating a larger number of event frames compared to static data to accumulate sufficient information. However, conventional practices in event data classification tasks often bind the time steps of SNN with the number of event frames, hindering the low-latency requirements for event data classification.

To address this challenge, we propose disentangling the relation between the time steps of SNN and the number of event frames. Specifically, during the training stage, we ensure that SNN learns as much information from event frames as possible to enhance performance. Conversely, during the inference stage, we aim to minimize the utilization of event frames to reduce the time steps and consequently decrease latency in SNN.

In this paper, we propose a novel Hybrid Step-wise Distillation (HSD) method for event data classification tasks. We advocate for the partition of the SNN training process into a pre-training phase and a fine-tuning phase. The pre-training phase aims to facilitate the

acquisition of more event frame information by the SNN, thereby enhancing overall performance. In contrast, the fine-tuning phase is designed to ensure that the SNN meets the low-latency requirements during the inference stage. Consequently, we partition event data into two distinct parts.

Given the limited information in event data, the first part leverages the robust feature extraction capabilities of ANN to learn the spatial features of event data. Subsequently, employing ANN-SNN conversion, these features are transmitted to the SNN during the pre-training phase. The second part involves fine-tuning the converted SNN to capture the spatio-temporal characteristics of event data under low time steps as effectively as possible. However, owing to the scarcity of information in event data compared to static data, the average output distribution of SNN across all moments can be more susceptible to the influence of individual time steps with abnormal output distributions, particularly at lower time steps.

To address this challenge, we introduce the Step-wise Knowledge Distillation (SKD) module. This module facilitates the transfer of the "Soft Labels" learned by the ANN during the pre-training phase to the output distribution of each time step in the SNN. This mechanism ensures a more stable training process, mitigating the impact of individual abnormal time steps and enhancing the overall robustness of the network. Experimental results demonstrate that our method achieves a balance between accuracy and latency in classification tasks on neuromorphic datasets.

Our contributions can be summarized fourfold:

- **New Benchmark.** We present a novel HSD that disentangles the dependency between the number of event frames and the time steps of SNN, maximizing the utilization of rich event data information. To our knowledge, we are the first to utilize both ANN-SNN conversion and knowledge distillation in classification tasks on neuromorphic datasets.
- **New Training Strategy.** We overcome the limitations of vanilla knowledge distillation SNNs, which rely solely on averaging information. Our proposed SKD optimizes from the perspective of compressed temporal dimension, considering distribution variances at each time step.
- **New Test Setting.** We surpass the limitation of utilizing all event frames during the inference stage. Our innovative test setting requires partial event frames to strike a balance between accuracy and latency by partitioning the neuromorphic dataset.
- **Solid Verification.** Our results at lower time steps either match or outperform the performance of most methods at longer time steps on synthetic datasets, *e.g.*, CIFAR10-DVS, N-CALTECH101, and real-world DVS-GESTURE.

## 2 RELATED WORK

**ANN-SNN Conversion** involves replacing the rectified linear unit (ReLU) activation function in ANN with integrate-and-fire (IF) neurons, and mapping the performance of ANN to SNN through weight sharing. Aligning SNN performance closely with ANN often necessitates longer time steps to synchronize the spike firing rate of SNN with the activation values of ANN. Recent studies have endeavored to bridge this gap post-conversion. Rueckauer et al. [32] introduced "soft-reset" to mitigate information loss during IF

neuron firing. Deng and Gu [7] proposed to regulate the relation between ANN's activation values and SNN's spike firing rates. Bu et al. [3] introduced quantization clip-floor-shift (QCFS) activation function as a ReLU replacement, offering a closer approximation to the activation function of SNN. You et al. [45] proposed an ANN-SNN conversion framework called SDM for action recognition tasks, which effectively overcomes conversion errors. Departing from the conventional use of ANN-SNN conversion on static datasets, we adopt this method for ANN-SNN conversion to initialize SNNs on neuromorphic datasets.

**Direct-training SNN** addresses non-differentiable challenges by introducing surrogate gradients. Some researchers aim to enhance performance by improving representative capabilities and alleviating training complexities. Fang et al. [12] proposed learning membrane parameters with spiking neurons to enhance the expressiveness of SNNs. LIAF-SNN [38] utilized analog values to represent neural activations instead of traditional binary values in leaky integrate-and-fire (LIF) SNNs. EICIL [34] expanded the representation space of spiking neurons. Some studies advocate for enhancing SNN architecture through attention mechanisms, such as [40, 42, 47, 48]. Li et al. [24], Shen et al. [35] aimed to improve the generalization of SNNs by incorporating data augmentation techniques. In order to improve the generalizability of SNNs on event-based datasets, He et al. [17] used static images to assist SNN training on event data. Meng et al. [27] proposed SLTT method that can achieve high performance while greatly improving training efficiency compared with BPTT. Unlike ANN, SNN introduces a temporal dimension, prompting many studies to focus on this aspect. Threshold-dependent BatchNorm (tdBN) [46] extended temporal dimension information to conventional BatchNorm. Deng et al. [8] expanded common loss functions to the temporal dimension to enhance generalizability. Ding et al. [10] proposed SSNN alleviates the temporal redundancy of SNN and significantly reduces inference latency. And Li et al. [25] adjusted the number of time steps based on different samples to reduce the latency of SNN.

**Knowledge Distillation** is initially proposed by [19] for model compression, subsequently utilized to enhance the performance of student models by transferring knowledge from teacher models. Romero et al. [30] first introduced the feature transfer into knowledge distillation (KD) and subsequently improved it through a series of works [5, 18]. Recent research in SNNs, such as [21], employing large SNN teacher models to guide training of smaller SNN student models. Takuya et al. [36] introduced an ANN-teacher model to instruct SNN-student models. Xu et al. [41] utilized feature-based and logit-based information from ANNs to distill SNNs. Guo et al. [15] proposed a joint training framework of ANN and SNN, in which the ANN can guide the SNN's optimization. In contrast to compressing the temporal dimension information of SNNs as in their methods, we consider variations in the output distribution of SNNs at each time step.

## 3 PRELIMINARIES

### 3.1 Partition of Neuromorphic Datasets

Neuromorphic datasets typically utilize the Address Event Representation (AER) format, represented as $E(x_i, y_i, t_i', p_i)$ where $i$ ranges from 0 to $N-1$, to convey event location in the asynchronous stream, timestamp, and polarity. To manage the large volume of events, we aggregate them into frames for processing, following methodologies outlined in previous studies [12, 48]. Specifically, events are partitioned into $T$ slices, with events within each slice cumulatively accumulated. The integrated event at location $(x, y)$ in the $j$-th slice ($0 \leqslant j \leqslant T - 1$) is denoted as:

$$F(j, p, x, y) = \sum_{i=j_l}^{j_r - 1} \mathcal{L}_{p,x,y}(p_i, x_i, y_i), \quad (1)$$

where the function $\mathcal{L}_{p,x,y}(p_i, x_i, y_i)$ serves as an indicator. The indices $j_l$ and $j_r$ correspond to the minimal and maximal timestamp indexes within the $j$-th slice. It is worth to note that $T$ signifies the total number of time steps in the training stage under our experimental setup.

In HSD, the time steps $T$ also represents the total number of input event frames in the training stage. We partition it into two segments: the preceding segment consists of $T_1$ event frames, serving as input for pre-training phase, while the subsequent segment comprises $T_2$ event frames, utilized by SNN in the fine-tuning phase. Therefore, it can ensure that SNN learns complete event frame information during the training stage, improving accuracy, while the inference stage only uses partial event frames for inference, reducing latency. Specifically, $T = T_1 + T_2$.

### 3.2 Neuron Models

For ANNs, the input $a^{l-1}$ to layer $l$ undergoes a linear transformation using matrix $W^l$ and a nonlinear activation function $f(\cdot)$, where $l = \{1, 2, 3, \cdots, L\}$,

$$a^l = f\left(W^l a^{l-1}\right), \quad (2)$$

where $f(\cdot)$ typically chosen as ReLU activation function.

In SNNs, IF neuron model is commonly employed for converting ANNs to SNNs [2, 3, 7]. To minimize information loss during inference, our neurons perform a "reset-by-subtraction" mechanism [33], where the firing threshold $\theta^l$ is subtracted from the membrane potential upon firing. The fundamental kinetic equations of IF neuron can be expressed as:

$$v^l(t) = v^l(t - 1) + W^l s^{l-1}(t)\theta^{l-1} - s^l(t)\theta^l, \quad (3)$$

where $v^l(t)$ denotes the membrane potential of layer $l$ at the $t$-th time step. $W^l$ is the synaptic weight between layer $l$-1 and layer $l$, and $\theta^l$ is the spike firing threshold in the $l$-th layer. $s^l(t)$ represents whether the spike fires at time step $t$, which is defined as:

$$s^l(t) = H\left(u^l(t) - \theta^l\right), \quad (4)$$

where $u^l(t) = v^l(t - 1) + W^l s^{l-1}(t)\theta^{l-1}$ denotes the membrane potential of neurons before spiking at time step $t$, $H(\cdot)$ represents the Heaviside step function. The output spike $s^l(t)$ is set to 1 if the membrane potential $u^l(t)$ exceeds the threshold $\theta^l$ and 0 otherwise.

## 4 PROPOSED METHOD

Hybrid Step-wise Distillation (HSD) method utilizes a unified model integrating both ANN and SNN components. The ANN model

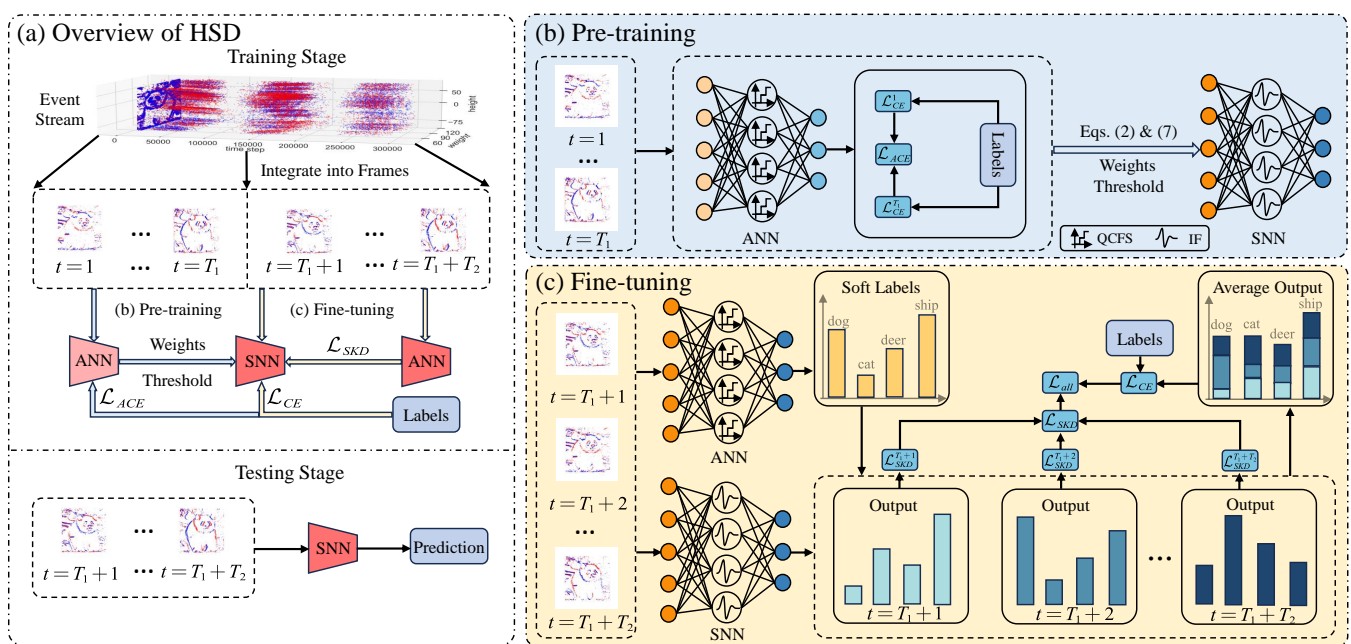

**Figure 3: (a) Overall framework of proposed HSD. It includes pre-training phase and fine-tuning phase. Initially, the raw event stream undergoes integrating to form event frames. Subsequently, the event frames from the neuromorphic dataset are partitioned into two segments. In the pre-training phase, an ANN processes $T_1$ event frames to transmit rich spatial information to SNN. In the fine-tuning phase, ANN provides learned "Soft Labels" guidance to influence SNN's output at each time step. (b)–(c) illustrate the details of the two phases of training.**

serves dual roles as the pre-training model and the teacher model, as illustrated in Fig. 3.

## 4.1 ANN Temporal Training

Considering that the event stream generates discrete events every 1 $\mu s$, and the information carried by individual events is limited, the conventional method is to aggregate the event stream into continuous event frames [12, 24, 37, 42, 46]. However, while the event stream contains temporal information at the millisecond level, the number of integrated event frames typically falls below 100, diminishing the significance of temporal information. Furthermore, for event data classification tasks, spatial information holds greater importance. Therefore, we opt not to employ a 3D network for processing event frames in the spatio-temporal training of ANN, but instead, we utilize a 2D network. This decision not only reduces computational overhead but also facilitates the migration of ANN-SNN conversion used for static datasets.

For ANNs, we utilize consecutive event frames, treating each frame as independent. Therefore, for ANN training, we employ the average cross-entropy loss:

$$\mathcal{L}_{\text{ACE}} = \frac{1}{T_1} \sum_{t=1}^{T_1} \mathcal{L}_{\text{CE}} \left( \boldsymbol{y}_t, \boldsymbol{y}_{\text{true}} \right), \quad (5)$$

where $\boldsymbol{y}_t$ denotes the model's prediction probability, $\boldsymbol{y}_{\text{true}}$ signifies the true labels, and $T_1$ represents the number of event frames utilized for ANN training. $\mathcal{L}_{\text{ACE}}$ resembles TET loss [8], tailored to improve the model's generalization.

## 4.2 ANN-SNN Conversion

To facilitate the conversion from ANN to SNN, aligning the firing rates (or postsynaptic potentials) of spiking neurons with ReLU activation outputs of artificial neurons is crucial. By integrating Eq. (3) from $t = 1$ to $T_{\text{AS}}$ and normalizing by $T_{\text{AS}}$, resulting:

$$\frac{\sum_{t=1}^{T_{\text{AS}}} \boldsymbol{s}^l (t) \theta^l}{T_{\text{AS}}} = \boldsymbol{W}^l \frac{\sum_{t=1}^{T_{\text{AS}}} \boldsymbol{s}^{l-1} (t) \theta^{l-1}}{T_{\text{AS}}} + \left( -\frac{\boldsymbol{v}^l (T_{\text{AS}}) - \boldsymbol{v}^l (0)}{T_{\text{AS}}} \right), \quad (6)$$

where $T_{\text{AS}}$ denotes the total simulation cycle of ANN-SNN conversion. For simplicity, we substitute the term $\frac{\sum_{t=1}^{T_{\text{AS}}} \boldsymbol{s}^l (t) \theta^l}{T_{\text{AS}}}$ in Eq. (6) with the average postsynaptic potential:

$$\boldsymbol{\phi}^l (T_{\text{AS}}) = \boldsymbol{W}^l \boldsymbol{\phi}^{l-1} (T_{\text{AS}}) + \left( -\frac{\boldsymbol{v}^l (T_{\text{AS}}) - \boldsymbol{v}^l (0)}{T_{\text{AS}}} \right), \quad (7)$$

where mirrors the forward propagation Eq. (2) in ANNs when $\boldsymbol{\phi}^l(T_{\text{AS}}) \geqslant 0$. This indicates that lossless ANN-SNN conversion can be attained as $T_{\text{AS}}$ methods infinity. For high-performance SNNs under low-latency, Bu et al. [3] suggested substituting the ReLU activation function in source ANNs with QCFS function:

$$\boldsymbol{a}^l = f\left( \boldsymbol{a}^{l-1} \right) = \frac{\lambda^l}{L} \text{clip} \left( \left\lfloor \frac{\boldsymbol{W}^l \boldsymbol{a}^{l-1} L}{\lambda^l} + \frac{1}{2} \right\rfloor, 0, L \right), \quad (8)$$

where $L$ denotes ANN quantization step, and $\lambda^l$ signifies the trainable threshold of the outputs in ANN layer $l$, which corresponds to the threshold $\theta^l$ in SNN layer $l$. Our work adheres to the conversion framework [3], employing QCFS function. Consequently,

the SNN inherits the rich spatial information acquired by the ANN, mitigating the risk of excessive information loss.

### 4.3 Step-wise Knowledge Distillation

Vanilla KD is categorized into feature-based distillation, logic-based distillation, and relation-based distillation [14]. We opt for logic-based distillation following [19]. Soft labels are preferred over hard labels as they allow student models to retain the predicted probability distribution obtained from teacher models. We employ Kullback-Leibler (KL) divergence to constrain the student's output to match the distribution of the teacher. Thus, the vanilla $\mathcal{L}_{\text{KD}}$ is defined as:

$$\mathcal{L}_{\text{KD}} = \sum_{i=1}^{N} \left( p_\tau^a(i) \log \frac{p_\tau^a(i)}{p_\tau^s(i)} \right), \tag{9}$$

where $p_\tau^a$ and $p_\tau^s$ denote the predicted distributions for ANNs and SNNs, respectively, and $N$ represents the total number of samples. During SNN training, the standard method integrates information across all time steps to derive the final prediction, yielding:

$$p_\tau^s = \frac{1}{T_2} \sum_{t=1}^{T_2} p_\tau^{s,t}, \tag{10}$$

where $T_2$ denotes the time steps of fine-tuning phase, and $p_\tau^a$ resembles Eq. (10). The output of the vanilla KD student model follows this pattern. However, since SNN outputs the probability distribution of each category at every time step, as $T_2$ increases, the final average probability distribution can roughly reflect the relation between each category. Nonetheless, when $T_2$ is small, the ability of the average probability distribution to represent the overall category probability distribution is limited. In contrast to SNNs on static datasets, the input at each time step for SNNs on neuromorphic datasets exhibits dynamic variability, leading to a significant difference in the distribution of the final output. To address this issue, SKD module is constructed to transfer the probability distribution learned by the ANN teacher model to each time step of SNN, facilitating a smoother distribution of output for SNN. Therefore $\mathcal{L}_{\text{SKD}}$ is defined as:

$$\mathcal{L}_{\text{SKD}} = \frac{1}{T_2} \sum_{t=1}^{T_2} \sum_{i=1}^{N} \left( p_\tau^a(i) \log \frac{p_\tau^a(i)}{p_\tau^{s,t}(i)} \right). \tag{11}$$

### 4.4 Training Framework

The overall training algorithm is outlined in Algorithm 1. To address the non-differentiability inherent in training SNNs via back-propagation, we employ the surrogate gradient technique. We choose triangular-shaped surrogate gradients [8], it can be described as:

$$\frac{\partial H(x)}{\partial x} = \frac{1}{\gamma^2} \max(0, \gamma - |x - V_{\text{th}}|), \tag{12}$$

where $\gamma$ is the constraint factor determining the sample range for activating the gradient, while $V_{\text{th}}$ denotes the threshold of IF neuron. We set $\gamma$ to 1 and $V_{\text{th}}$ to 1. The final loss function is expressed as:

$$\mathcal{L}_{\text{all}} = \mathcal{L}_{\text{CE}} + \lambda \mathcal{L}_{\text{SKD}}, \tag{13}$$

where $\mathcal{L}_{\text{CE}}$ denotes cross-entropy loss, while $\lambda$ signifies a hyperparameter to adjust the proportion of $\mathcal{L}_{\text{SKD}}$.

---

**Algorithm 1:** Hybrid Step-wise Distillation for Neuromorphic Dataset Classification.

**Require:** ANN model $f_{\text{ANN}}(x; W)$ with initial weights $W$; event frames $E_1$ and $E_2$, corresponding to quantities $T_1$ and $T_2$; quantization step $L$; initial dynamic thresholds $\lambda$; ANN-SNN conversion epochs $S_1$, and fine-tuning epochs $S_2$; fine-tuning time steps $T_2$

**Ensure:** SNN model

1 # Pre-training ANN-SNN
2 **for** $l = 1$ to $f_{\text{ANN}}$.layers **do**
3     Replace ReLU($x$) with QCFS($x$; $L, \lambda^l$)
4     Replace MaxPooling layer with AvgPooling layer
5 **end**
6 **for** $e = 1$ to $S_1$ **do**
7     **for** each event frame in $E_1$ **do**
8        Sample minibatch ($x^0$, $y$)
9        **for** $l = 1$ to $f_{\text{ANN}}$.layers **do**
10           Apply QCFS $x^l = $ QCFS($W^l x^{l-1}$; $L, \lambda^l$)
11        **end**
12     **end**
13 **end**
14 **for** $l = 1$ to $f_{\text{ANN}}$.layers **do**
15     Transfer weights to SNN $f_{\text{SNN}}.\hat{W}^l \leftarrow f_{\text{ANN}}.W^l$
16     Transfer threshold to SNN $f_{\text{SNN}}.\theta^l \leftarrow f_{\text{ANN}}.\lambda^l$
17     Set initial states to SNN $f_{\text{SNN}}.v^l(0) \leftarrow f_{\text{SNN}}.\theta^l/2$
18 **end**
19 # Fine-tuning SNN
20 **for** $e = 1$ to $S_2$ **do**
21     **for** each event frame in $E_2$ **do**
22        Sample minibatch ($x$, $y$)
23        **for** $t = 1$ to $T_2$ **do**
24           Compute prediction $y_{\text{ANN}}^{\text{pre}} = $ ANN($x$), $y_{\text{SNN}}^{\text{pre},t} = $ SNN($x$)
25           Compute distillation loss $\mathcal{L}_{\text{SKD}}^t = $ KL($y_{\text{ANN}}^{\text{pre}}, y_{\text{SNN}}^{\text{pre},t}$)
26        **end**
27        Aggregate loss over time $\mathcal{L}_{\text{SKD}} = \frac{1}{T_2} \sum_{t=1}^{T_2} \mathcal{L}_{\text{SKD}}^t$
28     **end**
29 **end**
30 **return** SNN model

---

## 5 EXPERIMENTAL RESULTS

### 5.1 Experimental Settings

**Datasets.** CIFAR10-DVS [22] dataset includes 10k dynamic vision sensor (DVS) images, adapted from the original CIFAR10 dataset. A training-validation split of 9:1 is used, resulting in 9k training images and 1k validation images. Initially, images are of $128 \times 128$ pixels but are resized to $48 \times 48$ for training, with the event data distributed across 10 frames per sample [8].

N-Caltech101 [28] dataset consists of 8,831 DVS images, sourced from the original Caltech101 dataset. The preprocessing method mirrors that of CIFAR10-DVS.

DVS-Gesture [1] dataset obtained using a DVS128 camera, it encompasses recordings of 11 hand gestures performed by 29 subjects under varying lighting conditions. This dataset organizes each event data into 16 frames.

**Metric.** Across all datasets, we utilize Top-1 accuracy to evaluate the model's performance [8].

**Implementation Details.** Our experimental setup leverages an NVIDIA Tesla V100 GPU, utilizing the PyTorch framework and the SpikingJelly package [11]. We employ the VGG-SNN [8] for CIFAR10-DVS and N-Caltech101, and SNN-5 [12] for DVS-Gesture. Data augmentation [24] are implemented consistently across all datasets. During the pre-training phase, the batch size is

**Table 1: Comparisons of Top-1 accuracy (%) performances with state-of-the-art methods on CIFAR10-DVS, N-Caltech101, and DVS-Gesture. $T$ denotes the time steps of SNN during the inference stage. † indicates reproduced results. Bold numbers are the best results.**

| Method | Venue | Model | CIFAR10-DVS | | N-Caltech101 | | DVS-Gesture | |
|---|---|---|---|---|---|---|---|---|
| | | | $T\downarrow$ | Acc $\uparrow$ | $T\downarrow$ | Acc $\uparrow$ | $T\downarrow$ | Acc $\uparrow$ |
| STBP-tdBN [46] | AAAI '21 | ResNet-19/17 | 10 | 67.80 | - | - | 40 | 96.87 |
| TA-SNN [42] | ICCV '21 | SNN-5/3 | 10 | 72.00 | - | - | 60 | 98.60 |
| PLIF [12] | ICCV '21 | SNN-4/5 | 20 | 74.80 | - | - | 20 | 97.60 |
| Dspike [23] | NeurIPS '21 | ResNet-18 | 10 | 75.40 | - | - | - | - |
| LIAF [38] | TNNLS '22 | LIAF-Net | 10 | 70.40 | - | - | 60 | 97.56 |
| NDA [24] | ECCV '22 | VGG-SNN | 10 | 79.60 | 10 | 78.20 | - | - |
| TCJA [48] | arXiv '22 | VGG-SNN | 10 | 80.70 | 14 | 78.50 | 20 | 99.00 |
| TET [8] | ICLR '22 | VGG-SNN | 10 | 83.17 | - | - | - | - |
| SpikFormer [47] | ICLR '23 | SpikFormer-4/2-256 | 10 | 78.90 | - | - | 10 | 96.90 |
| EventMixer [35] | Inf. Sci. '23 | ResNet-18 | 10 | 81.45 | 10 | 79.47 | 20 | 96.75 |
| SLTT [27] | ICCV '23 | VGG-11 | 10 | 82.20 | - | - | 20 | 97.92 |
| DT-SNN [25] | DAC '23 | VGG-16 | 5.25 | 74.40 | - | - | - | - |
| SSNN [10] | AAAI '24 | VGG-9 | 5 | 73.63 | 5 | 77.97 | 5 | 90.74 |
| TET [8] † | ICLR '22 | VGG-SNN/SNN-5 | 5 | 78.80 | 5 | 79.65 | 5/10 | 94.44/96.88 |
| TCJA [48] † | arXiv '22 | VGG-SNN/SNN-5 | 5 | 79.73 | 5 | 75.61 | 5/10 | 92.25/97.56 |
| SLTT [27] † | ICCV '23 | VGG-SNN/SNN-5 | 5 | 76.12 | 5 | 77.57 | 5/10 | 94.12/96.23 |
| HSD (Ours) | - | VGG-SNN/SNN-5 | 5 | **81.10** | 5 | **80.20** | 5/10 | **97.57/97.92** |

set to 128 for CIFAR10-DVS and N-Caltech101, and we utilize the stochastic gradient descent (SGD) [29] optimizer with a learning rate of 0.1. The batch size is set to 32 for DVS-Gesture, we employ the adaptive moment (Adam) estimation [20] optimizer with a learning rate of 1e-3. Training epochs is set to 300, adopting a cosine learning rate schedule and weight decay parameter of 5e-4. During the fine-tuning phase, we utilize the Adam optimizer with a learning rate of 1e-5, and training epochs is set to 100.

Specific to our method, the initial 5 frames are dedicated to pre-training ANN-SNN, while the subsequent 5 frames are reserved for fine-tuning the converted SNN on CIFAR10-DVS. A similar partitioning strategy is applied to N-Caltech101. Notably, owing to the richer temporal information on DVS-Gesture, the initial 6 frames are utilized for pre-training, followed by 10 frames for fine-tuning. Here, $T$ signifies the number of event frames involved in inference process, serving as the time steps for SNN, denoted as $T_2$ within the method.

## 5.2 Comparison with State-of-the-art Methods

In Tab. 1, we compare our experimental results with previous works. Concerning CIFAR10-DVS, our HSD has demonstrated superior accuracy compared to the majority of existing methods. At low time steps, our HSD achieves an accuracy of 81.10%, which is 7.47% higher than that of SSNN [10]. Although it may not attain the performance level of TET [8], it is noteworthy that our time step is only half of theirs. Moreover, when both time steps are set to 5, HSD surpasses TET with an enhancement of 2.30%. Additionally, we also reproduce the results of TCJA [48] and SLTT [27] on CIFAR10-DVS. Specifically, in $T$ = 5, TCJA achieves an accuracy of 79.73%, whereas our HSD outperforms TCJA with an improvement of 1.37%. And compared to SLTT, the performance improves by 4.98%.

Limited results are available for N-Caltech101. When the time step is 5, our HSD achieves an accuracy of 80.20%, exceeding SSNN by 2.23%. And, our HSD surpasses all baseline methods, even outperforming TET by 0.55% in $T$ = 5. Compared to TCJA and SLTT,

**Table 2: Comparisons of Top-1 accuracy (%) performances with different variants of HSD in $T$ = 5 on CIFAR10-DVS and N-Caltech101, and in $T$ = 10 on DVS-Gesture.**

| HT | KD | SKD | CIFAR10-DVS | N-Caltech101 | DVS-Gesture |
|---|---|---|---|---|---|
| ○ | ○ | ○ | 73.60 | 73.58 | 94.10 |
| ● | ○ | ○ | 79.21 | 79.97 | 95.83 |
| ○ | ○ | ● | 75.70 | 75.97 | 94.44 |
| ● | ○ | ● | **81.10** | **80.20** | **97.92** |
| ● | ● | ○ | 79.50 | 79.70 | 97.22 |
| ● | ○ | ● | **81.10** | **80.20** | **97.92** |

our HSD demonstrates a significant performance improvement of 4.59% and 2.63%.

On DVS-Gesture, we achieve competitive accuracy results of 97.92% with 10 time steps, surpassing the 97.60% accuracy of PLIF [12] and the 96.90% accuracy of SpikFormer [47]. Under the same experimental settings, our method shows an increase of 1.04% compared to TET. Furthermore, under the condition of $T$ = 10, our method outperforms TCJA and SLTT. The limited performance improvement observed on DVS-Gesture can be attributed to the insufficient data and the fact that temporal aggregation with $T$ = 10 has already accumulated a substantial amount of spatio-temporal information. And when the time step is set to 5, HSD shows a significant performance improvement compared to TET, TCJA, and SLTT, reaching 97.57%. The main reason is that our proposed partition of neuromorphic datasets idea solves the problem of reducing training samples at low time steps.

## 5.3 Ablation Study

In our HSD, the proposed hybrid training (HT) module and SKD module have significantly enhanced the classification performance at lower time steps. Therefore, in Tab. 2, we aim to validate the effectiveness of these two modules on CIFAR10-DVS, N-Caltech101, and DVS-Gesture.

**Table 3: Comparisons of Top-1 accuracy (%) performances with different quantization steps in $T$ = 5 on CIFAR10-DVS, N-Caltech101, and in $T$ = 10 on DVS-Gesture.**

| Quantization Step | 4 | 8 | 16 | 32 |
|---|---|---|---|---|
| CIFAR10-DVS | 77.23 | 80.12 | **81.10** | 76.77 |
| N-Caltech101 | 76.24 | 77.78 | 78.77 | **80.20** |
| DVS-Gesture | 94.79 | 95.48 | **97.92** | 96.18 |

*5.3.1 HT and SKD.* We observe that our baseline model VGG-SNN [8] achieves an accuracy of 73.60%. The introduction of HT module enables SNN to capture more event frame information within lower time steps, resulting in a notable 5.61% performance improvement, reaching an accuracy of 79.21%. SKD module effectively transfers the "Soft Labels" acquired by ANN to SNN, leading to a 2.10% enhancement and an accuracy level of 75.70%. When both HT and SKD modules are employed, there is a 7.50% increase in Top-1 accuracy. Similar improvements in accuracy can also be observed on N-Caltech101 and DVS-Gesture. However, on DVS-Gesture, the improvement with HT module is not substantial, as the SNN has already accumulated a sufficient amount of event frame information in $T$ = 10.

*5.3.2 SKD and KD.* To verify the effectiveness of the SKD module at low time steps. We compare the accuracy of vanilla KD with that of SKD across different datasets. SKD, by considering the temporal dimension of SNN and the variations in output distributions at different time steps, has demonstrated greater effectiveness in transferring knowledge from ANN to SNN compared to vanilla KD. This enhanced knowledge transfer results in improved SNN performance, especially within lower time steps.

In Tab. 2, compared to vanilla KD, SKD module shows performance enhancements on all three datasets. Notably, for CIFAR10-DVS, we observe a 1.60% increase in accuracy. In addition, for N-Caltech101 and DVS-Gesture experience significant accuracy improvements of 0.50% and 0.70%, respectively.

## 5.4 Effect and Selection of Quantization Step $L$

In our HSD, the hyper-parameter quantization step $L$ in pre-training phase significantly influences the accuracy of the converted SNN. To assess the impact of $L$ and identify the optimal value, we train VGG-SNN [8] and SNN-5 [12] using QCFS activation with different quantization steps $L$, specifically 4, 8, 16, and 32. Subsequently, we convert these models into corresponding SNNs.

The experimental results are presented in Tab. 3. It is evident that employing an excessively small quantization step $L$ adversely impacts the model's capacity, leading to a decrease in the final classification accuracy. Conversely, when the quantization step $L$ is too large, it requires the use of lower time steps $T$ during the subsequent fine-tuning phase. Therefore, it leads to a decrease in performance. Thus, for CIFAR10-DVS, N-Caltech101, and DVS-Gesture, we select quantization steps $L$ of 16, 32, and 16, respectively.

## 5.5 Robustness with Lower Time Steps

The selection of time steps in SNNs significantly impacts both inference latency and power consumption. Achieving lower time steps is a critical goal for SNNs. In this work, we evaluate the robustness of HSD under reduced time steps and compare it with TET [8]. We evaluate the performance of VGG-SNN model trained on synthetic datasets, *e.g.*, CIFAR10-DVS and N-Caltech101. Our analysis indicate that when there is a discrepancy between training and inference time steps, the accuracy of SNNs declines rapidly. To underscore the superiority of our method, we maintain consistency in training and inference time steps within TET by ensuring they are equal, *i.e.*, TET (STS).

For CIFAR10-DVS, setting the time step to 1, TET (STS) achieves a performance of 74.82%, while our HSD attains a higher accuracy of 75.81%. As depicted in Fig. 2(a), HSD's performance consistently exceeds TET (STS)'s at each time step. And in Fig. 2(b), concerning N-Caltech101, HSD uniformly surpasses TET (STS) in accuracy across all time step, underscoring HSD's superior efficacy, even at extremely low time steps on synthetic datasets. However, for N-Caltech101, the improvement of accuracy from HSD is not as significant as CIFAR10-DVS, this is because N-Caltech101 dataset only has approximately 80 samples per class, so the improvement from HSD is limited.

In the case of DVS-Gesture, there is a notable distinction: training and inference stages involve different time steps. Remarkably, when $T$ is greater than or equal to 2, its accuracy surpasses scenarios where training and inference stages share the same time steps, as depicted in Fig. 2(c). This is primarily due to the relatively small size of DVS-Gesture, which comprises only 1, 342 samples. Compared to the other two datasets, this dataset is characterized by limited data, and using more training data during the training stage may assist the model in better generalizing to testing data, thereby enhancing overall performance. For consistency in comparison, the same time steps are maintained for both the training and inference stages on DVS-Gesture.

To evaluate the performance of our HSD on real-world DVS-Gesture, we compare it with TET on SNN-5 [12]. While TET may not yield the best results on DVS-Gesture, it accounts for the variations at each time step and maintains the overall network architecture, aligning with our HSD in these respects. As depicted in Fig. 2(c), in $T$ = 1, TET (STS) achieves an accuracy of 90.97%. This is mainly due to the relatively small size of DVS-Gesture. Our HSD also achieves promising results, reaching 91.32%, indicating a marginal improvement of 0.35% over TET (STS). With the increase in $T$, both methods exhibit significant accuracy enhancements, with HSD consistently outperforming TET (STS) and TET (DTS). Overall, leveraging a larger number of event frames and integrating an ANN as a teacher model, our HSD surpasses TET (STS) and TET (DTS), particularly at extremely low time steps.

## 5.6 Visualization

To understand more clearly the feature extraction capability of HSD and the spatio-temporal characteristics of SNN, we visualize feature maps at different time steps $T$ in the first spiking neurons layer [12] on CIFAR10-DVS, specifically $T$ = 1 and 5. In Fig. 4(a)–(d), it is evident that as $T$ increases, the texture constructed by firing rates becomes more distinct. Our method leverages a larger amount of event frame information and benefits from guidance provided by the teacher model ANN, resulting in feature maps with more pronounced characteristics compared to those of TET [8].

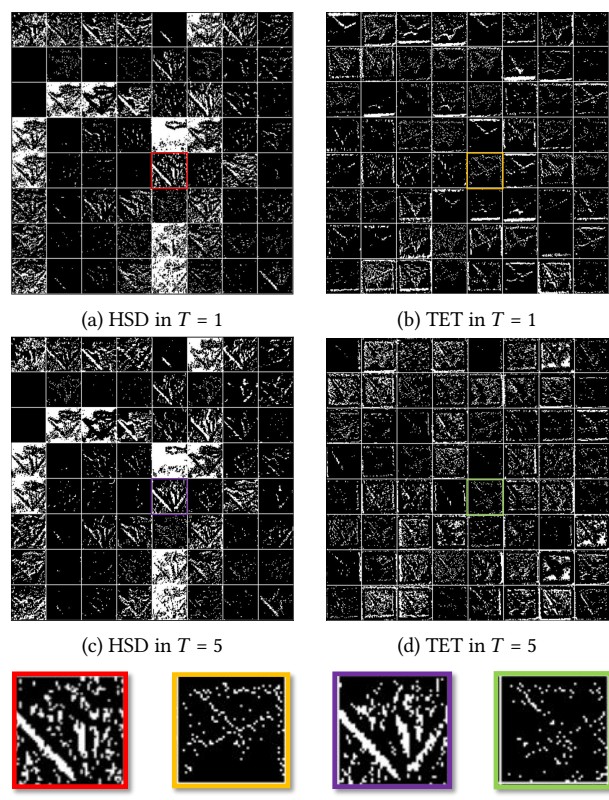

(a) HSD in $T = 1$     (b) TET in $T = 1$

(c) HSD in $T = 5$     (d) TET in $T = 5$

(e) channel 37

**Figure 4: Feature visualization for the initial spiking encoder. (a)–(d) depicts HSD and TET [8] in $T = 1$ and 5 on CIFAR10-DVS. (e) provides the corresponding feature visualizations for channel 37.**

**Table 4: Comparisons of Top-1 accuracy (%) performances with/without HT module in $T = 5$ on CIFAR10-DVS and N-Caltech101, and in $T = 10$ on DVS-Gesture.**

| Method | CIFAR10-DVS | N-Caltech101 | DVS-Gesture |
|---|---|---|---|
| Baseline | 73.60 | 73.58 | 94.10 |
| SLTT [27] *w/o* HT | 75.46 | 76.03 | 95.92 |
| SLTT [27] *w* HT | 78.78 | 79.14 | 97.26 |
| TET [8] *w/o* HT | 77.60 | 77.23 | 96.23 |
| TET [8] *w* HT | 80.82 | 79.65 | 97.58 |
| SKD (Ours) *w* HT | **81.10** | **80.20** | **97.92** |

Furthermore, Fig. 4(e) specifically presents the features of channel 37, revealing consistent patterns in its performance.

### 5.7 Generalization of Hybrid Training module

To verify the generalization of the proposed HT module, we consider applying it to TET [8] and SLTT [27]. We train a VGG-SNN model on CIFAR10-DVS. And in the fine-tuning phase, TET and SLTT are used instead of SKD module. In Tab. 4, as TET does not specifically consider the relation between the time steps of SNN and the number of event frames in classification tasks on neuromorphic datasets, the performance improvement is limited at low time

**Table 5: Comparisons of Top-1 accuracy (%) performances with different distillation modes in $T = 5$ on CIFAR10-DVS. $T = 5$ indicates that both the training and inference time steps of SNN are set to 5.**

| Response-based | Feature-based | Acc |
|---|---|---|
| ○ | ○ | 74.80 |
| ● | ○ | 76.40 |
| ○ | ● | 76.80 |
| ● | ● | 78.40 |

steps. However, when combined with our proposed HT module to learn more event frame information, we find that the performance improves by 3.22%, reaching 80.82%. However, it still falls short of HSD, mainly because HSD utilizes ANN as the teacher network and its powerful feature extraction ability. Furthermore, the performance improvement of SLTT on CIFAR10-DVS is similar to that of TET. And similar performance enhancements can be observed on N-Caltech101 and DVS-Gesture.

### 5.8 Comparison with the Other ANN Works

In Tab. 1, we compare our HSD with other works and illustrate the advantages of our work. Furthermore, we consider that vanilla KD can improve the performance of student models. So we choose [41] and conduct the experiment on CIFAR10-DVS. Among them, ensuring that the training and inference time steps of SNN are consistent. As shown in Tab. 5, in response based KD, the soft labels retains the hidden information in the output of the teacher model compared to the hard labels, resulting in a 1.60% improvement compared to the baseline. Moreover, feature-based knowledge utilizes the hidden information from the intermediate layers of ANNs to actively guide the training process of SNNs. Hence, the performance improves by 2.00%, reaching 76.80%. When combined with these two KD methods, achieving an accuracy of 78.40% and improving by 3.60%. This is because it helps the student model SNN obtain richer intermediate layer features and output category probabilities from the teacher model ANN. But its performance is still lower than our HSD, because HSD can enable SNN to learn as much event frame information as possible during the training stage, while using fewer event frames to reduce the latency during the inference stage.

### 6 CONCLUSION

In this paper, we address the issue of performance degradation in classification tasks on neuromorphic datasets when operating at lower time steps. We propose Hybrid Step-wise Distillation (HSD) method, which addresses the dependency between the number of event frames and the time steps of SNN. This method involves partitioning the SNN training process into pre-training and fine-tuning phases. Additionally, we introduce Step-wise Knowledge Distillation (SKD) module to stabilize the output distribution of the SNN at each time step, thereby achieving a balance between accuracy and latency in classification tasks on neuromorphic datasets.

In future work, we aim to explore the potential of heterogeneous ANN and SNN architectures in classification tasks on neuromorphic datasets. Our goal is to achieve a better balance among accuracy, latency, and efficiency, ultimately realizing efficient event-based visual recognition.

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
