# OpenReview forum: "Towards Low-latency Event-based Visual Recognition with Hybrid Step-wise Distillation Spiking Neural Networks"
_acmmm.org/ACMMM/2024/Conference — MM2024 Poster_

### Official Review · Reviewer_hKED · 2024-05-06

**Rating:** 5
**Confidence:** 4

**Summary:**

Spiking neural networks are well suited for neuromorphic datasets due to their rich spatio-temporal information processing capabilities and event-driven properties. In this paper, the authors propose a hybrid stepwise distillation (HSD) approach to untangle the dependency between the number of event frames and the time step of SNNs, i.e., using more event frames in the training phase to improve the performance and fewer event frames in the inference phase to reduce the latency.

**Strengths:**

1. I think the biggest contribution of this paper is to unravel the relationship between the time step of SNNs and the number of event frames. In my opinion, this idea is presented for the first time, and previous work has only focused on improving the performance of SNNs on neuromorphic datasets without considering the latency of the model in the inference phase.
2. how to achieve the trade-off between high accuracy and low latency in event-based visual recognition? This issue has been discussed in the paper of static data [1, 2], although this paper's application to a neuromorphic dataset is indeed the first of its kind.
3. the experimental part is adequate and convincing.


[1]: Li, C., Jones, E. G., & Furber, S. (2023). Unleashing the Potential of Spiking Neural Networks with Dynamic Confidence. In *Proceedings of the IEEE/CVF International Conference on Computer Vision* (pp. 13350-13360).

[2]: Li, Y., Geller, T., Kim, Y., & Panda, P. (2024). SEENN: Towards Temporal Spiking Early Exit Neural Networks. *Advances in Neural Information Processing Systems*, *36*.

**Limitations:**

I see no significant weaknesses in this paper and I am inclined to accept the paper.

Nevertheless, I have two questions for the authors to answer:
1. if the event frame is divided into two segments, i.e.  $t=1,\cdots, t=T_1$ and $t=T_1+1,\cdots,t=T_1+T_2$. In which segment is the pre-trained ANN conducted? Or is it the whole event frames? I can't figure this out in Figure 3 and Algorithm 1.
2. In the current method, the fine-tuned SNN perform inference in the second segment, i.e., $t=T_1+1,\cdots,t=T_1+T_2$. Can the time steps chosen for the fine-tuning phase be any consecutive sequence? For example, from $t=T_1-1, \cdots, t=T_1+T_2-3$. Would different choices of time steps affect the experimental results?

**Suitability:**

2

---

### Official Review · Reviewer_mXLY · 2024-05-15

**Rating:** 3
**Confidence:** 3

**Summary:**

This paper proposes the HSD method to hybridize the ANN and SNN to improve the performance of SNN for event visual recognition. To improve the performance of the converted SNN, the authors used the ANN to distill the SNN to facilitate training. The experiments conducted by the authors on three datasets confirm the effectiveness of the proposed method.

**Strengths:**

This paper decouple the dependence of training event frames and inference timesteps by introducing ANNs and larger timesteps during training and using smaller timesteps during inference. Experiments show that this method is able to recognize event data with low latency, even when the timestep is 1.

**Limitations:**

1. The motivation for using an ANN to guide an SNN for event data recognition should be elaborated. It is generally agreed that the ANN to SNN method is only able to match or exceed the performance of ANN on static datasets, while it is difficult to achieve satisfactory performance on event data because both ANNs and converted SNNs are less capable of temporal feature extraction. Using ANNs to recognize event data on a frame-by-frame basis and guide SNNs through conversion and distillation is hardly convincing.
2. In this paper, the first T1 event frames are used to train the ANN, while the last T2 event frames are used for fine-tuning the SNN and testing. Using the later T2 event frames for testing wastes the earlier T1 event frames because the test still has to process those T1 event frames, undermining the low-latency advantage of this method. In addition, why didn't the authors use the first T2 event frames for testing, so as not to cause additional test-time processing overhead.
3. The experimental results are not convincing. Table 2 shows that the CIFAR10-DVS baseline achieves 73.60% accuracy, which exceeds the accuracy of tdBN, TA-SNN, LIAF, DTSNN, and SSNN with more time steps. Given this, I think the performance of this paper comes mainly from small tricks in the experimental setup such as data sizing, data augmentation, etc., which has different practices in the SNN community. The authors should report the differences in their experimental setups when making comparisons and try to ensure a fair comparison.

**Suitability:**

2

---

### Official Review · Reviewer_dpNn · 2024-05-24

**Rating:** 4
**Confidence:** 3

**Summary:**

This paper introduces a Hybrid Step-wise Distillation (HSD) method for low-latency event-driven visual recognition using Spiking Neural Networks (SNNs). The approach leverages the temporal dynamics of SNNs to efficiently process spike events, resulting in a significant reduction in recognition latency.

**Strengths:**

1. By incorporating a Step-wise Knowledge Distillation (SKD) module, the issue of anomalous outputs at low temporal steps is addressed, enhancing the performance of the Spiking Neural Networks (SNNs).
2. The method achieves exceptional performance on neuromorphic dataset classification tasks, particularly at low temporal steps.

**Limitations:**

1. The ANN-SNN conversion method achieves high accuracy but experiences noticeable latency during the inference stage.
2. The reductions in latency are primarily observable in traditional static data scenarios, but in event-based vision tasks, high accuracy has not been achieved at extremely low time steps

**Suitability:**

3

---

### Official Review · Reviewer_rs6b · 2024-05-28

**Rating:** 6
**Confidence:** 3

**Summary:**

This paper proposes an innovative approach that uses fewer event frames during the inference stage to reduce latency. Additionally, it presents a new testing framework using partial event frames to balance accuracy and latency. Empirical evidence demonstrates that this method achieves competitive performance.

**Strengths:**

1.	This paper proposes an innovative training strategy that integrates advancements from both ANNs and SNNs.
2.	It presents a new testing framework that supports future research in the SNNs field.
3.	The experiment results are solid and the paper is easy to read.

**Limitations:**

1.	Some professional words lack explanation, such as DTS and STS.
2.	How to obtain soft labels? Why aren't soft labels applied during the pretraining stage?
3.	In the experimental section, what is the ``response-based’’? It should be described in details.

**Suitability:**

2

---

### Meta-Review · Area_Chair_QWh9 · 2024-06-27

**Recommendation:** Accept (Poster)
**Confidence:** 5

**Metareview:**

The paper proposes an innovative Hybrid Step-wise Distillation (HSD) method for low-latency event-driven visual recognition using Spiking Neural Networks (SNNs). The reviewers acknowledge the novelty and potential impact of the approach, highlighting its ability to balance accuracy and latency effectively. While there are some concerns about the explanation of technical terms, the rationale for using ANNs, and the fairness of the experimental setup, the overall consensus leans towards acceptance due to the solid empirical results and the significant contribution to the field. Therefore, I recommend accepting this submission.